# Could early tweet counts predict later citation counts? A gender study in Life Sciences and Biomedicine (2014–2016)

Tahereh Dehdarirad *

Department of Communication and Learning in Science, Chalmers University of Technology, Gothenburg, Sweden

* tahereh.dehdarirad@chalmers.se

## Abstract

In this study, it was investigated whether early tweets counts could differentially benefit female and male (first, last) authors in terms of the later citation counts received. The data for this study comprised 47,961 articles in the research area of Life Sciences & Biomedicine from 2014–2016, retrieved from Web of Science's *Medline*. For each article, the number of received citations per year was downloaded from WOS, while the number of received tweets per year was obtained from PlumX. Using the hurdle regression model, I compared the number of received citations by female and male (first, last) authored papers and then I investigated whether early tweet counts could predict the later citation counts received by female and male (first, last) authored papers. In the regression models, I controlled for several important factors that were investigated in previous research in relation to citation counts, gender or Altmetrics. These included journal impact (SNIP), number of authors, open access, research funding, topic of an article, international collaboration, lay summary, F1000 Score and mega journal. The findings showed that the percentage of papers with male authors in first or last authorship positions was higher than that for female authors. However, female first and last-authored papers had a small but significant citation advantage of 4.7% and 5.5% compared to male-authored papers. The findings also showed that irrespective of whether the factors were included in regression models or not, early tweet counts had a weak positive and significant association with the later citations counts (3.3%) and the probability of a paper being cited (21.1%). Regarding gender, the findings showed that when all variables were controlled, female (first, last) authored papers had a small citation advantage of 3.7% and 4.2% in comparison to the male authored papers for the same number of tweets.

## Introduction

According to statistics provided by the US National Science Foundation [1], women received over half of the bachelor's, master's and doctorate degrees awarded in biological sciences in 2016. Furthermore, the proportion of women amongst researchers in health and life sciences between 2011–2015 was shown to be overall higher than men researchers, as per the *Gender in*

**Data Availability Statement:** All relevant data regarding the variables, how they were obtained, and R packages used for analysis are within the paper. The results of multi-collinearity tests and data for Fig 2 (at aggregated level) are also available in the supporting information. The Doi list

of the documents used in this study are available at https://doi.org/10.7910/DVN/GHMV8Q in compliance with NLM Copyright. However, restrictions apply to the availability of the bibliometric and altmetrics (tweet counts) data. These data were downloaded under the provision of the institutional standard contract held by the Chalmers University of Technology to Clarivate's Web of Science (https://clarivate.com/webofsciencegroup/solutions/web-of-science/), SciVal (https://www.scival.com/) and PlumX (https://plu.mx/sign_in). The author did not have any special access privileges to these databases. Interested researchers may access Clarivate's Web of Science, PlumX and SciVal in the same way the author did.

**Funding:** The author(s) received no specific funding for this work.

**Competing interests:** The authors have declared that no competing interests exist.

the *Global Research Landscape Report* [2]. Despite the gender parity in degree recipients and the number of researchers, women remain underrepresented among tenure-track biomedical faculty at research institutions [3] and are underrepresented in the faculties of medicine and life sciences, as well in senior positions [4]. In 2016 in the EU, women represented totally 27% of grade A academic staff in health and medical sciences [5]. In the United States, in 2014, women composed 38% of the full-time academic medicine workforce, while men made up 62%.Additionallly, only 21% of full professors and just 15% of department chairs were female, compared with 79% and 85% for men in the same positions [6].

Beyond the gender imbalance in the number of women in senior positions and amongst tenure-track faculty members, some studies have also reported a citation advantage for male-last authored papers in biomedical and life sciences fields [7–9]. However, some others did not find notable differences between male and female-last authored papers or between female and male authors in terms of the number received citations [10, 11]. Typically, in biomedicine, the author listed first has less experience and does most of the research work, while the author listed last has more experience and provides a supervisory role [12, 13].

Given these gender differences in the number of women in senior positions and the number of citations, some studies have also sought to examine whether the web might provide a democratizing space for female academics [14, 15]. Many social media sites have provided new opportunities for both female and male scholars to disseminate and promote their research results within and beyond scientific community [15, 16]. Twitter is one of these social media platforms which is fundamentally reshaping the way biomedical scientists and academic physicians can discover, discuss and share research across disciplinary boundaries, as well as to the public. Twitter allows conversations about new papers to happen immediately and publicly [15, 17]. It also provides a possibility for authors to push their research out via twitter, rather than hope that it is pulled in by readers. Thus, this possibility provides the potential for scholars to draw wide attention to their research [15]. Twitter might also reduce the influence of hierarchies based on seniority. This is because on Twitter, people who do not have tenure, or have a limited number of publications or are early in their career can demonstrate their expertise [17]. Shifting to the push method on social media might also potentially reduce gendered gatekeeping in the dissemination of research [15]. For example, some studies on social media and gender have found that females had a higher visibility in terms of Web citations [18], average Mendeley readers [19, 20], profile views on Academia.Edu in certain disciplines [21], or event counts from Twitter [14, 20], blogs, and news [14]. Others found similar visibility for both female and male scholars in blogs, news, Facebook, or LinkedIn. [20].

Regarding the relation between tweet counts and the number received citation, studies generally tend to suggest a weak positive correlation [15, 22]. Some studies also suggested that tweet counts could predict the later number of received citation [23] or correlate with later downloads and citations for arXiv preprints [24].

This study follows in the same vein. However, it extends this line of research about the prediction of later citation count by early tweets, by comparing female and male (first, last) authored papers while controlling for several important factors that according to previous research have an association with citation counts. Most of these factors have also been examined in relation to gender or altmetrics studies. In relation to citation and gender, these included factors such as journal impact [7, 25], citations and self-citation of first and last authors [25], topic of an article (measured as MeSH) [8, 26], number of MeSH topics [13], gender of first and last authors [9], and the total number of authors' publications [13, 27]. In relation to citations, these included open access (OA) [28], Mega journal [29, 30] and number of topics of an article [31] amongst others. Regarding altmetrics and citations, factors such as abstract readability [32], international collaboration (measured as number of countries) [32,

33], title length [34], OA [35, 36], Mega journal, journal impact (measured as SNIP) and lay summary [36] have been studied.

Additionally, by studying both first and last authorship positions, I was able to control for the effect of seniority in terms of citations and tweet counts.

Thus, this study aims to investigate whether early tweets counts could differentially benefit female and male scholars in the field of Life Sciences and Biomedicine, in terms of later citations counts received per paper. The study has the two following objectives:

1. To compare the number of received citations by female and male last/first authored papers, when controlling for a number of important factors.

2. To investigate whether (and to what extent) early tweet counts can predict the later number of received citations by female and male last/first authored papers, when controlling for several important factors.

## Materials and methods

### Data collection and processing

The data for this study comprised 47,961 articles in the research area of Life Sciences & Biomedicine from 2014–2016, retrieved from Web of Science's *Medline* in June 2020. The reason for choosing this time period was to give the documents the required two-year period after publication year to receive tweets, which could be used to predict later citation counts. Furthermore, it would ensure that the documents would have had enough time (a time citation window of at least three years) to be cited following the two-year period after publication year. For each article, the number of received citations per year was downloaded from WOS, while the number of received tweets per year was downloaded from PlumX, using a combination of Doi and Pubmed ID. As the number of tweets per year is not currently available in PlumX, I obtained the date of tweets for each article and then aggregated the number of citations and tweets per year using the methodology applied in Thelwall and Nevill's study [37] (See Table 1).

After aggregation, of the 47,961 articles, 2,496 had zero citations and 24,190 had zero tweets. Fig 1 shows the distribution of early tweet counts versus later citation counts.

OA status of the articles was obtained from Unpaywall.org in June 2020. To determine the gender of first and last authors, Gender API (https://gender-api.com/) was used. Using this service, it is possible to search for first names, including those with two parts. The results provide the gender (male, female, or unknown), the number of names used to determine the gender and accuracy [38]. In cases of gender-neutral, unknown, initials or where the accuracy was lower than 80%, the names were checked manually using internet searches. The gender of authors in 12 authorship positions were remained unidentified. In the regression models (explained in data collection processing section) they were regarded as missing values. These 12 authorships accounted for seven first and five last authorship positions. The reason for choosing these two authorship positions was that in the field of biomedicine, the last position in the authors list is reserved for senior authors, whereas the first author position is for the

**Table 1. The aggregation method for early tweet counts and later citation counts for each year.**

| Year | Number of articles | Early number of tweets | Later citation counts |
|------|--------------------|-----------------------|-----------------------|
| 2014 | 16,630 | Sum of tweets counts for 2014 and 2015 | Sum of citations coutns for 2016–2020 |
| 2015 | 16,404 | Sum of tweets counts for 2015 and 2016 | Sum of citations coutns for 2017–2020 |
| 2016 | 14,927 | Sum of tweets counts for 2016 and 2017 | Sum of citations coutns for 2018–2020 |

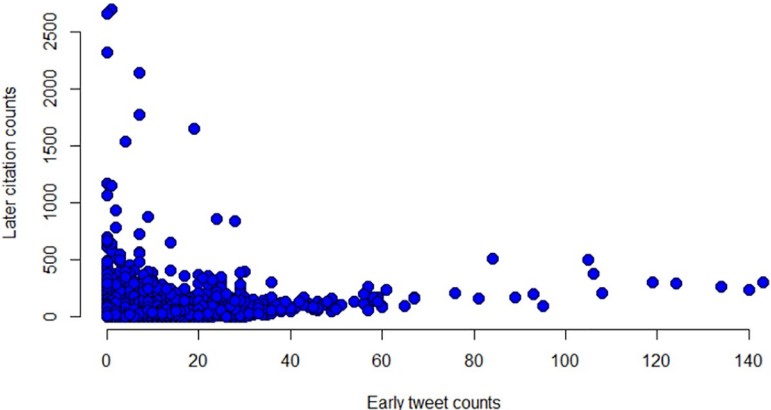

**Fig 1. The scatter plot of early tweet and later citation counts.**

person who fulfils the International Committee of Medical Journal Editors (ICMJE) authorship criteria to the highest level and performs the majority of the experimental and clinical work [12, 39].

The total numbers of publications, citations and self-citations for each author (first, last) as scientific (professional) age of an author [13, 25] were downloaded from SciVal API using authors' IDs in June 2020. To do this, the author IDs for first and last authors were downloaded via Scopus author API using a combination of Doi and Pubmed ID of articles. Then, in the next step, the SciVal Author Lookup API (https://dev.elsevier.com/documentation/SciValAuthorAPI.wadl) was used to download the three former-mentioned indicators.

Regarding mega journals, I used the journal list provided by Spezi et al.'s study [40] to determine whether a journal was mega journal or not. A mega journal is a peer-reviewed academic open access journal that publishes manuscripts that presents scientifically trustworthy empirical results without asking about the potential scientific contribution prior to publication. It covers a broad coverage of different subject areas and uses article processing charges to cover the costs of publishing [29, 40].

For each article, the paper length was considered as the absolute number of pages of a publication, while the title length was calculated by counting the number of characters in the title of an article.

As this study has been conducted in the area of Life Sciences and Biomedicine, MeSH categories were an appropriate subject classification to consider. Medline assigns articles to 14 broad MeSH categories. In this article, only the seven most relevant medical topics were considered for evaluation. These topic categories were Anatomy, Organisms, Diseases, 'Chemicals and Drugs', 'Analytical, Diagnostic and Therapeutic Techniques and Equipment', 'Psychiatry and Psychology' and 'Health Care'. For each of these seven MeSH categories, a dummy variable was created and entered as a covariate in the regression models. The articles that did not belong to any of these seven MeSH categories were tagged with a 0.

Lay summaries can help journals in life sciences and biomedicine to reach out to patients and others who might benefit from the research [41]. Thus, they may assist the diffusion of research on social media platforms such as Twitter. Using a journal list provided by Shailes [41], the articles were divided in two groups, those with and those without a lay summary.

Regarding the abstract readability, the Flesch Reading Ease Score was used, as it is the most commonly used measure of text readability and it has been used in other bibliometric studies [30]. The R quanteda package was used to calculate this score for each abstract. The highest

possible score is 121.22 and there is no lower limit. Very complicated sentences can have negative scores. The higher the score, the easier the text is to understand.

F1000 score as an altmetric indicator was included in the models as a control variable. The rationale for this is that the articles scored in F1000 are recommended as highly important works in the fields of life sciences, health and physical sciences [42].

Table 2 shows all the variables studied in this paper categorized as dependent, independent variables and covariates. It also provides a short description of these variables and how they are measured.

**Table 2. Dependent variables, independent variables and covariates for the hurdle models.**

| Variable type | Name | Measure |
|---|---|---|
| **Dependent** | Later citation counts[1] | The number of received citations after the two first years of publication |
| | Total number of citations[2] | The total number of citations received by an article since its publications. |
| **Independent and Covariate** | Early tweet counts | The number of tweets in the first two years of publication |
| | Gender | Gender of first and last author on an article: Male (0); Female (1) |
| | Number of authors | Number of authors collaborating in an article |
| | Funding | Funded article (1); not-funded article (0) |
| | SNIP | Source (journal) Normalized Impact per Paper. |
| | International collaboration | Number of countries collaborating in an article. |
| | Number of MeSH topics | Number of MeSH headings assigned to an article |
| | MeSH category | Seven MeSH categories assigned to each article as listed below:<br>MeSH1: Anatomy (1); Otherwise[3] (0)<br>MeSH2: Organisms (1); Otherwise (0)<br>MeSH3: Diseases (1); Otherwise (0)<br>MeSH4: Chemicals and Drugs (1); Otherwise (0)<br>MeSH5: Analytical, Diagnostic and Therapeutic Techniques and Equipment (1); Otherwise (0)<br>MeSH6: Psychiatry and Psychology (1); Otherwise (0)<br>MeSH7: Health Care (1); Otherwise (0) |
| | Title length | Number of characters in the title of an article. |
| | Lay summary | Articles from journals including lay summaries listed in Shailes list[4] (1); other journals (0) |
| | Abstract readability | Flesch readability score of the abstract. |
| | F1000 score | The score was obtained from Altmetrics.com public API. |
| | Mega journal | Mega Journal (1); non- Mega journal (0) |
| | OA | OA (1); non-OA articles (0) |
| | Paper length | The absolute number of pages of a publication derived from the beginning and end page of a document. |
| | Total number of publications, self-citations and citations for first and last authors | These values were downloaded from SciVal API for first and last authors using their authors' IDs. |

1. Dependant variable in the tweet-citation regression analysis (Models 2, 3).

2. Dependent variable in citation analysis (Model 1).

3. By otherwise I mean the other 13 MeSH categories.

4. https://elifesciences.org/articles/25411?utm_source=content_alert&utm_medium=email&utm_content=fulltext&utm_campaign=elife-alerts.

## Data analysis and procedures

Excel and the *mctest*, *pscl*, *quanteda* R packages were used to process and analyse the data.

Considering that the dependent variable of this study, the number of citations (See Table 2), was count data, count regression models were used. Furthermore, as this variable was over-dispersed and zero-inflated, a count model was required to deal with these two issues. Therefore, a negative binomial-logit hurdle model was the best fit for the data. Hurdle models measure the likelihood of an observation being positive or zero, and then determine the parameters of the count distribution for positive observations. Thus, a hurdle model comprises two parts: the count model, which is either a negative binomial or Poisson model, and the logit model. The count model predicts the changes in the positive non-zero observations, whilst the logit part models the zero observations [43, 44].

In this paper, three hurdle regression analysis were performed, namely model 1, model 2 and model 3. In model 1, the total number of received citations was considered as a dependant variable, whereas the gender of first and last authors was considered as independent variables. The rest of the variables were considered as covariates, except time since publication, which was considered as an offset variable in the regression model.

In models 2 and 3, the later citation counts were considered as dependant variables, whereas the early tweet counts were considered as an independent variable. In Model 3, the rest of variables were considered as covariates. In model 2, there were no covariates. In both models, time after two first years of publication was entered as an offset variable. Table 3 shows descriptive statistics at paper level for the covariates used in regression models 1 and 3. As can be seen from this table, the covariates are divided into two groups of numerical and categorical variables.

**Table 3. Descriptive statistics at paper level for numerical and categorical covariates entered in the regression models 1 and 3.**

| Numerical Covariates | Mean (SD) | Median | Categorical Covariates | Category | Number (%) |
|---|---|---|---|---|---|
| Title length | 12.87 (4.64) | 12 | Mega journal | Yes | 4,597 (9.58) |
| Number of Mesh topics | 5.91 (1.51) | 6 | | No | 43,364 (90.42) |
| SNIP | 1.33 (0.97) | 1.11 | OA Status | OA | 27,906 (58.18) |
| Number of authors | 5.38 (6.82) | 4 | | non-OA | 20,055 (41.82) |
| Number of countries | 1.48 (0.99) | 1 | Lay summary | Yes | 1,580 (3.29) |
| Abstract readability | 14.04 (13.44) | 14.47 | | No | 46,381 (96.71) |
| F1000 score | 0.04 (0.34) | 0.001 | Funding | Yes | 10,289 (21.45) |
| Paper length | 9.87 (26.19) | 9 | | No | 37,672 (78.55) |
| Total publications for last authorship position | 70.39 (74.69) | 49 | MeSH category | Anatomy | 3,942 (8.22) |
| Total number of citations for last authorship position | 2456 (4825.87) | 1093 | | Organisms | 5,351 (11.16) |
| Total number of self-citations for last authorship position | 183.50 (361.80) | 86 | | Diseases | 3,244 (6.76) |
| Total publications for first authorship position | 22.82 (33.50) | 13 | | Chemicals and Drugs | 3,431 (7.15) |
| Total number of citations for first authorship position | 682.24 (1922.98) | 212 | | Analytical, Diagnostic and Therapeutic Techniques and Equipment | 11,863 (24.73) |
| Total number of self-citations for first authorship position | 48.87 (146.69) | 12 | | Psychiatry and Psychology | 814 (1.70) |
| | | | | Health Care | 1,651(3.44) |

Multicollinearity was tested using Variance Inflation Factor (VIF). The VIF estimates how much the variance of a regression coefficient is inflated due to multicollinearity in the model. As a rule of thumb, a VIF value that exceeds 5 or 10 indicates a problematic amount of collinearity [45]. All variables had VIF values less than 3; hence no collinearity is expected (See S1 and S2 Appendices).

## Results

Fig 2 shows the percentage of first and last authorship positions by gender. As can be seen from the figure, in both authorship positions, the percentage of male-authored papers is higher than that of female authors. However, the percentage of female-first authored positions is slightly higher than the percentage of female-last authored positions.

### Comparison of the number of received citations between female and male last/first authored papers (Model 1)

As can be seen from the count model in Table 4, female first-authored papers and female-last authored papers have a small significant citation advantage over male-authored papers in these two positions. This means that by a unit of increase in gender (moving from male to female in either of these positions), the average number of received citations will be increased by 4.7% for female first-authored papers and by 5.5% for female last-authored papers.

As for controlled factors in both logit and count models and regardless of gender, OA, SNIP, number of authors, paper length and the total number of citations and self-citations of first and last authors has a positive associations with average number of received citations, as well as higher probability of a paper to be cited. Amongst MeSH topics, there was small positive association with the estimated number of received citations for articles categorized as 'Chemicals and Drugs'.

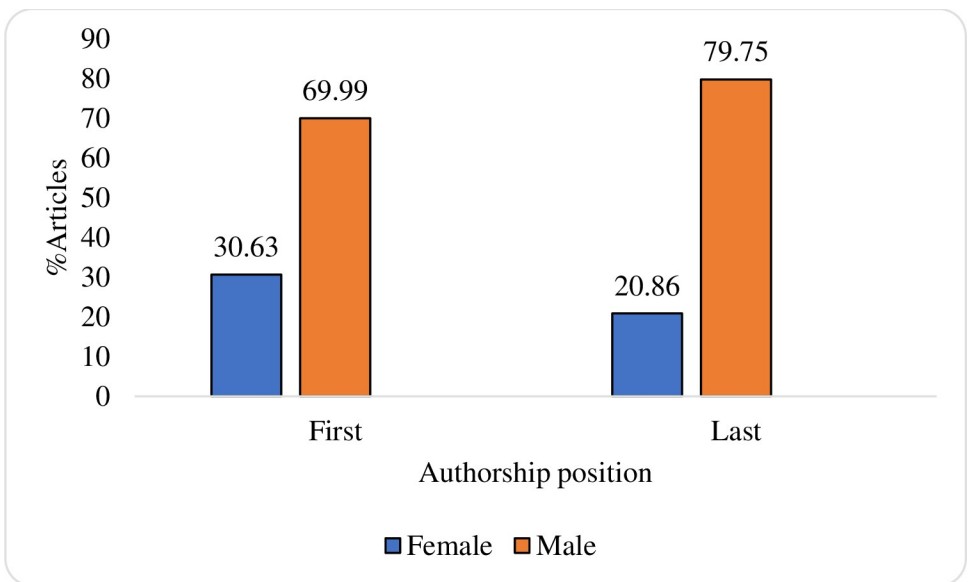

**Fig 2. The percentage of first and last authored papers by gender in the life Sciences and Biomedicine (2014–2016).**

**Table 4. Results of Hurdle regression for the comparison of female and male first/last authored articles regarding citation counts, having controlled for several factors.**

| Count model | Total number of citations | | |
|---|---|---|---|
| | Coef. | Exp(Coef.) | Sig. |
| Last (Female) | 0.054 | 1.055 | < 0.001*** |
| First (Female) | 0.045 | 1.047 | < 0.001*** |
| Title length | -0.008 | 0.992 | < 0.001*** |
| Mega Journal | -0.038 | 0.963 | 0.545 |
| Number of MeSH topics | -0.006 | 0.994 | 0.112 |
| MeSH-Anatomy | 0.035 | 1.035 | 0.078 . |
| MeSH-Organism | 0.006 | 1.007 | 0.704 |
| MeSH-Diseases | 0.036 | 1.036 | 0.093 . |
| MeSH-Chemicals and Drugs | 0.046 | 1.048 | 0.021 * |
| MeSH-Analytical. Diagnostic and Therapeutic Techniques and Equipment | 0.015 | 1.015 | 0.274 |
| MeSH-Psychiatry and Psychology | 0.035 | 1.036 | 0.394 |
| MeSH-Health Care | -0.085 | 0.918 | 0.004 ** |
| SNIP | 0.309 | 1.363 | < 0.001*** |
| OA | 0.216 | 1.242 | < 0.001*** |
| Number of authors | 0.010 | 1.010 | < 0.001 *** |
| International collaboration | 0.023 | 1.024 | < 0.001*** |
| Lay Summary | 0.040 | 1.041 | 0.582 |
| F1000 | 0.108 | 1.115 | < 0.001 *** |
| Funding | -0.024 | 0.976 | 0.081 . |
| Paper length | 0.008 | 1.008 | < 0.001 *** |
| Abstract readability | -0.001 | 0.999 | 0.007 ** |
| Total publications of last author | -0.001 | 0.998 | < 0.001 *** |
| Total number of citations for last author | 0.162 | 1.177 | < 0.001 *** |
| Total number of self-citations for last author | 0.0001 | 1.000 | < 0.001 *** |
| Total publications of first author | -0.006 | 0.993 | < 0.001 *** |
| Total number of citations for first author | 0.309 | 1.362 | < 0.001 *** |
| Total number of self-citations for first author | 0.0006 | 1.001 | < 0.001 *** |
| Logit model | Total number of citations | | |
| | Coef. | Exp(Coef.) | Sig. |
| Last (Female) | 0.144 | 1.155 | 0.051 . |
| First (Female) | 0.060 | 1.062 | 0.347 |
| Title length | 0.020 | 1.020 | 0.002 ** |
| Mega Journal | 0.315 | 1.370 | 0.493 |
| Number of MeSH topics | 0.032 | 1.032 | 0.115 |

(Continued)

**Table 4.** (Continued)

| Count model | Total number of citations | | |
|---|---|---|---|
| | *Coef.* | *Exp(Coef.)* | *Sig.* |
| MeSH-Anatomy | 0.004 | 1.004 | 0.974 |
| MeSH-Organism | 0.080 | 1.084 | 0.412 |
| MeSH-Diseases | 0.150 | 1.162 | 0.227 |
| MeSH-Chemicals and Drugs | 0.142 | 1.153 | 0.248 |
| MeSH-Analytical. Diagnostic and Therapeutic Techniques and Equipment | 0.027 | 1.028 | 0.713 |
| MeSH-Psychiatry and Psychology | -0.079 | 0.924 | 0.725 |
| MeSH-Health Care | -0.161 | 0.851 | 0.281 |
| SNIP | 1.020 | 2.774 | < 0.001 *** |
| OA | 0.176 | 1.192 | 0.010 * |
| Number of authors | 0.105 | 1.111 | < 0.001 *** |
| International collaboration | 0.060 | 1.062 | 0.261 |
| Lay Summary | -0.312 | 0.732 | 0.597 |
| F1000 | 0.281 | 1.325 | 0.371 |
| Funding | 0.240 | 1.271 | 0.021 * |
| Paper length | 0.035 | 1.036 | < 0.001*** |
| Abstract readability | -0.003 | 0.997 | 0.171 |
| Total publications of last author | -0.005 | 0.995 | < 0.001 *** |
| Total number of citations for last author | 0.144 | 1.155 | < 0.001*** |
| Total number of self-citations for last author | 0.001 | 1.001 | < 0.001 *** |
| Total publications of first author | -0.008 | 0.992 | < 0.001*** |
| Total number of citations for first author | 0.299 | 1.348 | < 0.001 *** |
| Total number of self-citations for first author | 0.002 | 1.002 | 0.004 ** |

Signif. codes: 0 '***' 0.001 '**' 0.01 '*' 0.05 '.' 0.1 ' ' 1.

## Early tweet counts and later citation counts

**Regression model with tweet and citation counts only (Model 2).** As can be seen from Table 5, the early number of tweets received by an article has a small positive association with the average number of later citations and a higher probability of a paper being cited. In other words, by increase of a unit in the number of tweet counts, the average number of later citation counts will approximately increase by 1.7% and the probability of being cited will be higher by 22.5%.

**Table 5. Results of Hurdle regression for the association between early tweet counts and later citation counts, without any control variables.**

| Early tweet Counts | Later citation counts | | | | | |
|---|---|---|---|---|---|---|
| | Count model | | | Logit model | | |
| | *Coef.* | *Exp(Coef.)* | *Sig.* | *Coef.* | *Exp(Coef.)* | *Sig.* |
| | 0.017 | 1.017 | < 0.001 *** | 0.203 | 1.225 | < 0.001 *** |

Signif. codes: 0 '***'.

**Regression model with tweet and citation counts controlling for all covariates (Model 3).** In the next step, we checked to see if there still would be an association between the early number of tweets and the later number of received citations when controlling for a several important factors that have an association with the number of received citations. Interestingly, the results from the comparison of two regression models (2 and 3) shows that by increase of a unit in early tweet counts in both models, the estimated average citation counts will increase to 1.7% in Model 2 (see Table 5) and 3.3% in the current model (see Table 6). As the coefficient in Model 3 is still significant and positive, this suggests that the early number of tweets could predict the later number of received citations, having controlled for specific factors.

As can been seen from count and logit models in Table 6, by increase of a unit in the number of early tweets, the estimated number of received citations will increase by 3.3% and the probability of being cited will be higher by 21.1%.

Regarding gender, as can be seen from the count model, it can be concluded that while keeping all other variables in the model constant, at the same number tweets, the estimated number of citations by female first or last-authored papers on average is slightly higher than that for male authors. In other words, by switching from male to female in both authorship positions, the average number of citations by female first or last-authored papers (at the same number of tweets for both genders) will increase by 3.7% and 4.2%, respectively.

As for the rest of covariates, as can be seen from both logit and count models, OA, SNIP, number of authors, International collaboration, Funding, paper length and being categorized under the 'Chemicals and Drugs' MeSH topic, were significantly associated with the higher number of received citations as well as the higher probability of being cited.

## Conclusion and discussion

The goal of this paper was to examine whether and to what extent early tweet counts received by articles in the field of Life Sciences and Biomedicine (2014–2016) could differently benefit female and male scholars in terms of the later citation counts received. To do this, the number of received citations by female and male last/first authored papers were compared, when controlling for several important factors (model 1). Then, it was investigated whether, and to what extent, early tweet counts could predict later citation counts received by female and male last/first authored papers, when controlling for several important factors (model 3). The findings in relation to these two objectives are briefly discussed below.

Regarding the first objective, the findings showed that the percentage of papers with male authors in first or last authorship positions was higher than that for female authors. Furthermore, the percentage of female-first authored papers was slightly higher than the percentage of female-last authored ones. These findings might indicate male dominance in the field. The later finding might especially reflect the lack of senior females in this field, as last authors tend to be senior. This conclusion is supported by Plank-Bazinet's, et al. [4] study which found a significant scarcity of women in academic biomedical leadership and senior positions. Having controlled for several factors, it was found that female first and last-authored papers had a small but significant citation advantage of 4.7% and 5.5% compared to male-authored papers. This finding is interesting given the lower number of female authors in these two authorship positions. The finding regarding the female first author citation advantage is in line with Thelwall's [9] study, which found a small citation advantage for female first authored papers in parasitology. As first authorship positions tend to be taken by younger researchers, it could be suggested that young female researchers are slightly outperforming young male researchers in terms of citation counts. The findings regarding the last authorship position, contradicts the ones from Thelwall [9], which found a small female last author disadvantage in immunology,

**Table 6. Results of Hurdle regression for the association between early tweet counts and later citation counts, having controlled for several factors.**

| Count model | Later citation counts | | |
|---|---|---|---|
| | *Coef.* | *Exp(Coef.)* | *Sig.* |
| Early tweet counts | 0.032 | 1,033 | < 0.001 *** |
| Last (Female) | 0.041 | 1,042 | 0.003 ** |
| First (Female) | 0.036 | 1,037 | 0.003 ** |
| Title length | -0.012 | 0.987 | < 0.001 *** |
| Mega Journal | -0.022 | 0.978 | 0.746 |
| Number of MeSH topics | -0.016 | 0.984 | < 0.001 *** |
| MeSH-Anatomy | 0.051 | 1.053 | 0.018 * |
| MeSH-Organism | 0.002 | 1.002 | 0.906 |
| MeSH-Diseases | 0.016 | 1.016 | 0.498 |
| MeSH-Chemicals and Drugs | 0.075 | 1.078 | <0.001 *** |
| MeSH-Analytical. Diagnostic and Therapeutic Techniques and Equipment | 0.020 | 1.021 | 0.179 |
| MeSH-Psychiatry and Psychology | 0.047 | 1.049 | 0.307 |
| MeSH-Health Care | -0.115 | 0.891 | < 0.001 *** |
| SNIP | 0.376 | 1.456 | < 0.001 *** |
| OA | 0.294 | 1.342 | < 0.001 *** |
| Number of authors | 0.011 | 1.011 | < 0.001 *** |
| International collaboration | 0.045 | 1.046 | < 0.001 *** |
| Lay Summary | 0.055 | 1.056 | 0.504 |
| F1000 | 0.121 | 1.129 | < 0.001 *** |
| Funding | 0.064 | 1.066 | < 0.001 *** |
| Paper length | 0.012 | 1.012 | < 0.001 *** |
| Abstract readability | -0.003 | 0.997 | < 0.001 *** |
| Total publications of last author | -0.000 | 1.000 | 0.015 * |
| Total number of citations for last author | 0.000 | 1.000 | < 0.001 *** |
| Total number of self-citations for last author | 0.000 | 1.000 | < 0.001 *** |
| Total publications of first author | -0.003 | 0.997 | < 0.001 *** |
| Total number of citations for first author | 0.000 | 1.000 | < 0.001 *** |
| Total number of self-citations for first author | 0.000 | 1.000 | <0.001 *** |
| Logit model | Later citation counts | | |
| | *Coef.* | *Exp(Coef.))* | *Sig.* |
| Tweet counts | 0.192 | 1.212 | < 0.001 *** |

*(Continued)*

**Table 6.** (Continued)

| Count model | Later citation counts | | |
|---|---|---|---|
| | *Coef.* | *Exp(Coef.)* | *Sig.* |
| Last (Female) | 0.107 | 1.113 | 0.095 . |
| First (Female) | 0.073 | 1.075 | 0.198 |
| Title length | 0.020 | 1.020 | < 0.001 *** |
| Mega Journal | 0.557 | 1.746 | 0.225 |
| Number of MeSH topics | 0.019 | 1.019 | 0.276 |
| MeSH-Anatomy | 0.092 | 1.096 | 0.361 |
| MeSH-Organism | 0.127 | 1.136 | 0.142 |
| MeSH-Diseases | 0.178 | 1.194 | 0.102 |
| MeSH-Chemicals and Drugs | 0.218 | 1.243 | 0.041 * |
| MeSH-Analytical. Diagnostic and Therapeutic Techniques and Equipment | 0.033 | 1.033 | 0.614 |
| MeSH-Psychiatry and Psychology | -0.156 | 0.855 | 0.417 |
| MeSH-Health Care | -0.131 | 0.877 | 0.319 |
| SNIP | 1.017 | 2.765 | < 0.001 *** |
| OA | 0.266 | 1.305 | < 0.001 *** |
| Number of authors | 0.110 | 1.116 | < 0.001 *** |
| International collaboration | 0.115 | 1.122 | 0.014 * |
| Lay Summary | 0.161 | 1.174 | 0.785 |
| F1000 | 0.269 | 1.308 | 0.319 |
| Funding | 0.422 | 1.524 | < 0.001 *** |
| Paper length | 0.041 | 1.042 | < 0.001 *** |
| Abstract readability | -0.004 | 0.996 | 0.048 * |
| Total publications of last author | -0.002 | 0.998 | 0.013 * |
| Total number of citations for last author | 0.000 | 1.000 | 0.711 |
| Total number of self-citations for last author | 0.002 | 1.002 | 0.007 ** |
| Total publications of first author | -0.004 | 0.996 | 0.007 ** |
| Total number of citations for first author | 0.000 | 1.000 | 0.437 |
| Total number of self-citations for first author | 0.004 | 1.004 | < 0.001 *** |

Signif. codes: 0 '***' 0.001 '**' 0.01 '*' 0.05 '.' 0.1 ' ' 1.

parasitology and virology. It is however in line with a study by Sotudeh, Dehdarirad, Freer [20] in the field of neurosurgery, which found a higher average of citations for female first and last authors.

Regarding the second objective, the findings showed that irrespective of whether the factors were included in regression models or not, early tweet counts had a weak positive and significant association with the later citations counts (3.3%) and the probability of a paper being cited (21.1%). This finding is in line with the ones of Eysenbach [23], Haustein, et al. [22], Peoples et al. [46] and Klar, et al. [15].

Regarding gender, the findings showed that while keeping all other variables constant in the model, at the same number of tweets, the average citation counts by female first or last-authored papers was slightly higher than that for male authors. Compared to male first or last

authored papers, female authored papers had a small citation advantage of 3.7% and 4.2% when both genders receive the same number of tweets per paper. This might suggest that in Life sciences and Biomedicine, early tweets counts could slightly benefit female authored papers in terms of the later citation counts received. This finding is to some extent in line with Klar et al. [15], who found a positive association between the percentage of women authors per paper and the number of citations received, after controlling for the number of tweets.

With regard to the other variables controlled for in model 3 (early tweet-later citation counts), the results showed that while keeping all other variables in the model constant, with the same number of tweets, two conclusions can be drawn. i) OA articles, articles with international collaboration or research funding had a higher average of citations and a higher probability to be cited. ii) As F1000 score and journal impact (SNIP) increased, the average number of citations increased. Amongst these covariates, journal impact and OA had the highest association with the number of citations and the probability of being cited, respectively. The finding about journal impact is consistent with Andersen's et al. [25] study, which found journal prestige as a covariate that accounted for most of the small average citation differences between genders. The finding about OA might show the importance of making an article open, as this makes it more visible, and thus easier for Twitter users to access the full text of articles. This in turn might translate into more citations.

With regard to MeSH topics, the results showed that while keeping all other variables in the model constant and with the same number of tweets, the articles with 'Chemicals and Drugs' MeSH topic had a higher probability of being cited (24.3%) and a higher average of citations (7.8%) in comparison to the rest of articles with other 13 MeSH topics. According to a study by Bhattacharya, Srinivasan and Polgreen [47], tweeting about MeSH topics such as 'Chemicals and Drugs' leads to more engagement (in terms of number of re-tweets) on Twitter. More engagement on Twitter does not guaranty more citations. However, it might provide increased visibility for papers with this topic, which may also make them be seen more by the scientific community.

The findings also showed that while some factors had a positive association with the average number of citations received (model 1, citation comparison), they had a very weak or almost no association with the later citation counts received (model 3, early tweet-later citation counts). As an example, the total number of citations by first and last authors had a positive association with the probability of a paper being cited (34.8%; 15.5%) and the average number of citations received in model 1 (36.2%; 17.7%). However, the association between the same variables and the average later citation counts in model 3 was almost none and the coefficients were very close to zero. Collectively, this could suggest that at the same number of tweets, scientific impact of authors, measured as total number of citations, has almost no association with the probability of a paper being cited and later average citations counts received.

This study has some limitations. The extent to which early tweet counts associates with later citation counts may vary by adding or removing factors from the model. However, the current model attempted to control for several important factors. By doing so, I was able to increase the probability of obtaining a more precise and reliable association between the early tweet counts and average number of citations received. It also should be considered that the analysis in this paper was limited to articles in the area of Life sciences and Biomedicine which were published in the time period of 2014–2016. Thus, the results obtained in this article are not comprehensive. Thus, caution should be advised with generalization of the results beyond the case studied.

## Supporting information

**S1 Appendix. Multicollinearity diagnostics results for Model 1.**
(DOCX)

**S2 Appendix. Multicollinearity diagnostics results for Model 3.**
(DOCX)

**S1 Fig. The percentage and number of articles by female and male authors in first and last authorship positions.**
(XLSX)

## Acknowledgments

I would like to thank Mr. Jonathan Freer and Ms. Momena Khatun from the Department of Communication and Learning in Science (CLS) at Chalmers University of Technology for their assistance regarding the gender detection of authors. I also would like to thank Dr. Hajar Sotudeh from Shiraz University (Iran) for her valuable comments which helped to improve the paper. Finally, I would like to thank Mr. Alexander Mladenovic from Gothenburg University (Sweden) for his assistance regarding data collection.

## Author Contributions

**Conceptualization:** Tahereh Dehdarirad.

**Data curation:** Tahereh Dehdarirad.

**Formal analysis:** Tahereh Dehdarirad.

**Investigation:** Tahereh Dehdarirad.

**Methodology:** Tahereh Dehdarirad.

**Software:** Tahereh Dehdarirad.

**Validation:** Tahereh Dehdarirad.

**Visualization:** Tahereh Dehdarirad.

**Writing – original draft:** Tahereh Dehdarirad.

**Writing – review & editing:** Tahereh Dehdarirad.

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
