## [Decision Letter · Decision Letter 0]

24 Aug 2020

PONE-D-20-22052

Could early tweet counts predict later citation counts? A gender study in Life Sciences and Biomedicine (2014-2016)

PLOS ONE

Dear Dr. Dehdarirad,

Thank you for submitting your manuscript to PLOS ONE. After careful consideration, we feel that it has merit but does not fully meet PLOS ONE’s publication criteria as it currently stands. Therefore, we invite you to submit a revised version of the manuscript that addresses the points raised during the review process.

We look forward to receiving your revised manuscript.

Kind regards,

Alireza Abbasi

Academic Editor

PLOS ONE

Additional Editor Comments:

In addition to reviewers' comments, The following issues need to be investigates as well:

Several usage of ‘authors (first, last)’ OR ‘authors (last, first)’

- To avoid any confusion, please use a complete and proper term ‘first and last author’. That will help keeping consistency as well.

Please create a separate sub-section for the control variables discussed in pages 6 and 7. A proper (but brief) definition of each variable is also needed. no definition is provided for some variables (e.g. ‘mega journals’, ‘F1000’).

Some of the variables are publication-age dependent (e.g. ‘citations count’) and some not (e.g., ‘tweet counts’. In other words, the number of tweet counts are fixed to two years while ‘the number of citations’ are calculated regardless of the age of publication! That can affect the regression (and correlation) results. This can be addressed by for instance using a fixed number of years for citations count (a 3-year window, for instance) or using publication age as a factor. Please discuss.

Likewise, total number of publications, citations, … (last variable in Table 2) is age-dependent (i.e. older authors will have higher values) and can affect the statistical results. Please discuss.

The use of regression types should be justified.

Except for the number of articles per year, no statistic if provided for the data set. It will be helpful to provide some basis statistics about the dataset. For instance, total number of unique authors (and by gender, and position); number of articles and authors) with no citations / tweets; distribution of citations count, tweet count. A discussion on the effects of that range on the statistical analysis is also needed.

Under discussion for Multicollinearity test, it is claimed that there is no significant collinearity while a high correlation is expected between independent/covariate variables such as ‘number of authors’ and ‘international collaboration’. Please discuss.

Journal Requirements:

2) We note that you have indicated that data from this study are available upon request. PLOS only allows data to be available upon request if there are legal or ethical restrictions on sharing data publicly. For more information on unacceptable data access restrictions, please see http://journals.plos.org/plosone/s/data-availability#loc-unacceptable-data-access-restrictions.

Reviewers' comments:

Reviewer's Responses to Questions

**Comments to the Author**

1. Is the manuscript technically sound, and do the data support the conclusions?

Reviewer #1: Yes

Reviewer #2: Yes

2. Has the statistical analysis been performed appropriately and rigorously? 

Reviewer #1: Yes

Reviewer #2: I Don't Know

3. Have the authors made all data underlying the findings in their manuscript fully available?

Reviewer #1: No

Reviewer #2: Yes

4. Is the manuscript presented in an intelligible fashion and written in standard English?

Reviewer #1: Yes

Reviewer #2: Yes

5. Review Comments to the Author

Reviewer #1: This analysis adds to both the role of gender in citations and the relationship between altmetrics and citations. The methods are appropriate and careful. The discussion is also appropriate and careful. Although the regression *might* perhaps better have been done with ordinary least squares and log(1+citations) as the dependent variable, I think the approach used here with the hurdle aspect could be better so I do not recommend a change.

In Table 3, some of the commas should be full stops.

Line 304: "articles with International" should be "articles with international"

* I had to answer No to the question, "Have the authors made all data underlying the findings in their manuscript fully available?" but the author is correct not to share the citation and altmetric data.

Reviewer #2: The paper looks at early tweets (first two years) for papers in life sciences and citation counts and investigate whether there is an association between them and whether there are gender differences in this regard (citation advantage, and benefit from tweets). A relatively large number of papers have been studied which is good and regression analysis has been used for data analysis. The paper can benefit from some clarification in methods and presentation.

This sentence in the introduction, “but only 21% were full professors and just 15% were department chairs [6].” I think this is natural as academic rank is like a pyramid and there are fewer professors than associate professors and fewer associate professors than assistant professor. But if 21% of full professors were female (and the remaining 79% were male, and 15% of department chairs were female (and the remaining 85% were male) then that should be a concern. Not sure if this is what the author (and that reference) has meant to say?

Page 4 where it says “Most of these factors have also 91 been examined in relation to gender or altmetrics studies.”, and then lists several factors that have been studied, it should be made clear each of those factors was investigated in relation to what, gender or altmetrics. For instance, was the influence of abstract readability was studied in relation to altmetric or in relation to gender? This is important for understanding the contribution of the current paper.

The method needs more details and clarification. For instance, it says tweets for a two year period were collected. For example for papers published in 2014, tweets in 2014 and 2015 were collected. Was the month of publication taken into account in this data collection? If not, a paper published in January 2014 would’ve had two years of tweets in the dataset, while a paper in Dec 2014, would have only 13 months worth of tweets. The same goes for citation data.

How many authors (first and last) were there in the dataset and how the publications, citation, self-citation data was obtained? Did the author manually search each of those probably thousands of authors? Was there any problem with author disambiguation?

Title length: were words like the, a, an, on, and …counted?

Abstract readability, how was it calculated? Did software (text processing) do this or somebody had to read all of the abstracts and assign a score? How about the validity and reliability issues here?

Figure 1 should have proper legends with values shown on the bars (e.g. percentage).

The paper needs a table that presents some descriptive statistics about the variables included in the study. For instance, how many authors, how many papers from each subject category, what was the average and median title length, how many OA and non-OA, how many papers had funding and how many didn’t, average, mean of the number of authors etc.

I believe the level of accuracy used in the paper for significance reporting (shown with long exponents, e.g. 2.45e-05) is unnecessary, up to 3 decimal points would suffice.

Also I think the author needs to make the contribution clear in the paper given the focus is on association of tweets and citation (adding gender to the issue) and there has already been some good research on that.

Language, proofreading will improve the paper. It seems the paper has one author, but throughout the paper, the author uses 'we' to present the study which might not be right.

Typo: p. 209, line 206, as well as well as higher

Typo, p. 16, line 299, cations

6. PLOS authors have the option to publish the peer review history of their article (what does this mean?). If published, this will include your full peer review and any attached files.

Reviewer #1: **Yes: **Mike Thelwall

Reviewer #2: No

---

## [Author Response · Author response to Decision Letter 0]

5 Oct 2020

PONE-D-20-22052

Answer to editor’s comments

#Several usages of ‘authors (first, last)’ OR ‘authors (last, first)’- To avoid any confusion, please use a complete and proper term ‘first and last author’. That will help keeping consistency as well.

• Thanks for this comment. Following the editor’s comment, this has been corrected throughout the manuscript. 

# Please create a separate sub-section for the control variables discussed in pages 6 and 7. A proper (but brief) definition of each variable is also needed. no definition is provided for some variables (e.g. ‘mega journals’, ‘F1000’).

Regarding this comment, I believe it would be confusing to add a sub-section called control variables before explaining about the regression models and introducing the dependant and independent variables. The aim of data collection section was just to explain how the variables were collected and processed. Then, in data analysis section, I categorized them in different groups (independent, dependant and covariate) in relation to regression models and briefly explained how they are measured (Table 2). Then in Table (3) I have provided statistics for these variables.

Thus, instead I did as below: 

• Some of the variables were already explained in the manuscript such as F1000, abstract readability, paper length, abstract length and Mesh. However, now explanations regarding some variables such as Mega journals (page 7, line 161-165) and lay summary (page 8, lines 179-183) that were not explained in the manuscript have now been added. The following reference with regard to lay summary has been added to the reference section:

Shailes S. Plain-language Summaries of Research: Something for everyone. eLife. 2017; 6:e25411. doi: 10.7554/eLife.25411.

Furthermore, details regarding the data collection and the reason for inclusion of ‘the total numbers of publications, citations and self-citations’ has been also added to page 7, lines 153-159.

• I also added the following paragraph at the end of the data collection and processing section (page 9, lines 192-194), to refer the readers to more details about these variables such as type of variables (independent, dependant and covariates) and how they are measured. 

‘Table 2 shows all the variables studied in this paper categorized as dependent, independent variables and covariates. It also provides a short description for these variables and how they are measured.’ 

• Table 3 is now added to the manuscript which provides descriptive statistics and further details for these variables (page 11). 

If the editor still believes that the sub-section would be necessary, after these changes have been made, I will add it. 

#Some of the variables are publication-age dependent (e.g. ‘citations count’) and some not (e.g., ‘tweet counts’. In other words, the number of tweet counts are fixed to two years while ‘the number of citations’ are calculated regardless of the age of publication! That can affect the regression (and correlation) results. This can be addressed by for instance using a fixed number of years for citations count (a 3-year window, for instance) or using publication age as a factor. Please discuss.

• Regarding the number of tweets, a two-year time period after publications was set to be sure that articles have enough time to receive the number of tweets. The reason to this is that twitter is categorized as a fast altmetrics data sources. This means that for these sources, in this case Twitter, altmetric events for newly published research outputs are accumulated very fast. According to research by Feng and Costas (2020), half of the tweet mentions for newly published articles were accrued in the first 2 weeks (14 days) after the research outputs were published, and over 85% of their data happened within a year (365 days). Given that, by setting a two-year time period after publication, I made sure that the studied articles had enough time to receive tweets. I have already mentioned this on page 6 lines 121-122.

Fang, Z., Costas, R. Studying the accumulation velocity of altmetric data tracked by Altmetric.com. Scientometrics 123, 1077–1101 (2020). https://doi.org/10.1007/s11192-020-03405-9

• Regarding the number of citations, as already mentioned in page 6 lines 123-125, I set a citation window of at least 3 years for the studied articles to have enough time to receive citations. Furthermore, I controlled for the effect of time on the number of citations by entering the time after two first years of publication as an offset variable in the regression models (1,2,3). It is already mentioned in the manuscript at page 9 lines 213-214 and page 10, lines 217-218. So, using both these methods, I was able to control for the effect of time. 

#Likewise, total number of publications, citations, … (last variable in Table 2) is age-dependent (i.e. older authors will have higher values) and can affect the statistical results. Please discuss.

• The total number of publications, citations and self-citations of an author is defined as professional or scientific age of an author as per studies by Mishra et al. (2018) and Andersen et al. (2019). Following the methodology used in the above-mentioned studies, I entered the professional age in regression models (1,3) as control variables to be able to control for this age effect on the probable number of citations received or the probability of a paper being cited. I have now mentioned these studies in the manuscript at page 7 line 154. 

Article Source: Self-citation is the hallmark of productive authors, of any gender 

Mishra S, Fegley BD, Diesner J, Torvik VI (2018) Self-citation is the hallmark of productive authors, of any gender. PLOS ONE 13(9): e0195773. https://doi.org/10.1371/journal.pone.0195773

Andersen JP, Schneider JW, Jagsi R, Nielsen MW. Gender variations in citation distributions in medicine are very small and due to self-citation and journal prestige. eLife. 2019;8:e45374. doi: 10.7554/eLife.45374.

#The use of regression types should be justified.

• On pages 9, lines 201-209, it is already explained and justified why hurdle regression model has been used. 

#Except for the number of articles per year, no statistic if provided for the data set. It will be helpful to provide some basis statistics about the dataset. For instance, total number of unique authors (and by gender, and position); number of articles and authors) with no citations / tweets; distribution of citations count, tweet count. A discussion on the effects of that range on the statistical analysis is also needed.

Regarding this comment:

• It should be considered that the analysis in this paper is based on authorship positions and not by the number of publications for unique authors in each authorship position. In other words, the analysis in the paper is at paper level and not author level. Thus, providing statistics on author level would be confusing and is not in line with the aim of study. The data set of this study was comprised of 47,961 papers. So, there are 47,961 authorship positions for both last and first authorship position. The gender of authors in 12 authorship positions were not detected as mentioned on page 7 line 145. However, to be more precise, now, I have added this line to page 7, lines 147-148: ‘These 12 authorships accounted for seven first and five last authorship positions’. Thus, in this paper the statistics on the number of authorship positions by each gender is reported in Figure 2 at paper level. Additionally, Table 3 has now been added to the manuscript page 11, which provides statistics on the covariates. 

• Following the editor’s comment, the number of articles with zero citations and zero tweets are provided at paper level at Page 6, lines 134-135 as below. Fig 1. is also added to page 6, which shows the scatter plot for early tweet and later citation counts.

‘After aggregation, of the 47,961 articles, 2,496 had zero citations and 24,190 had zero tweets. Fig 1. shows the distribution of early tweet counts versus later citation counts.’ 

• Regarding the comment for the effect of distribution, in the methodology section when it explains about the hurdle regression and why it used (page 9, lines 201-209), the distribution of citations is considered. As explained in the manuscript (lines 201-209), one of the main reasons for using hurdle regression model in this paper is that the citations counts are over-dispersed and include an excessive number of zeros (zero-inflated). Furthermore, this model can handle zero values. Basically, the hurdle model has two parts: part one, deals with zero values and calculates the probability of receiving citations using a logit model. Part two, deals with non-zero values and predicts the changes in the positive non-zero observations, which in this paper is the probable number of received citations. The obtained results from both parts, zero values (logit model) and non-zero values (count model) are explained in results section and are discussed in discussion and conclusion section for each model in relation to each gender and to covariates. 

#Under discussion for Multicollinearity test, it is claimed that there is no significant collinearity while a high correlation is expected between independent/covariate variables such as ‘number of authors’ and ‘international collaboration’. Please discuss.

• Regarding multicollinearity test, I used mctest R package to test for multicollinearity amongst the variables. The results of this test for models 1 and 3 are shown in the Appendices (s1, s2). By looking at the VIF values in S1 and S2, it can be seen that the VIF values are all lower than 5. So, no collinearity was found. There is also explanation regarding this test and results on page 11 lines 226-230. 

• There are other studies in citation and altmetrics analysis like Zahedi and Haustein (2018), Didegah Bowman and Holmberg (2018), that have entered these covariates in the regression models as no-collinearity was found. 

1. Zahedi Z, Haustein S. On the relationships between bibliographic characteristics of scientific documents and citation and Mendeley readership counts: A large-scale analysis of Web of Science publications. Journal of Informetrics. 2018;12(1):191-202. doi: https://doi.org/10.1016/j.joi.2017.12.005.

2. Didegah F, Bowman TD, Holmberg K. On the differences between citations and altmetrics: An investigation of factors driving altmetrics versus citations for finnish articles. Journal of the Association for Information Science and Technology. 2018;69(6):832-43. doi: 10.1002/asi.23934.

Answer to Reviewers' comments

Reviewer #1: 

# This analysis adds to both the role of gender in citations and the relationship between altmetrics and citations. The methods are appropriate and careful. The discussion is also appropriate and careful. Although the regression *might* perhaps better have been done with ordinary least squares and log(1+citations) as the dependent variable, I think the approach used here with the hurdle aspect could be better, so I do not recommend a change.

• Thanks for this comment.

# In Table 3, some of the commas should be full stops.

• Thanks for this comment. It has now been corrected. Table 3 is now Table 4 in the manuscript.

# Line 304: "articles with International" should be "articles with international"

• It has now been corrected.

Reviewer #2: 

# This sentence in the introduction, “but only 21% were full professors and just 15% were department chairs [6].” I think this is natural as academic rank is like a pyramid and there are fewer professors than associate professors and fewer associate professors than assistant professor. But if 21% of full professors were female (and the remaining 79% were male, and 15% of department chairs were female (and the remaining 85% were male) then that should be a concern. Not sure if this is what the author (and that reference) has meant to say?

• Thanks for this comment. Yes, it is exactly what the report says. For details please refer to this pdf https://store.aamc.org/downloadable/download/sample/sample_id/228/, where it shows these percentages for men and women. I have now made this clearer in the manuscript (Page 3, lines 55-57).

#Page 4 where it says “Most of these factors have also 91 been examined in relation to gender or altmetrics studies.”, and then lists several factors that have been studied, it should be made clear each of those factors was investigated in relation to what, gender or altmetrics. For instance, was the influence of abstract readability was studied in relation to altmetric or in relation to gender? This is important for understanding the contribution of the current paper.

• Following the reviewer’s comment, the factors are related to each group of studies i.e. gender, altmetrics (pages 4-5, lines 93-101) have been added,

• Furthermore, the two following references have also been added to the manuscript:

35. Holmberg K, Hedman J, Bowman TD, Didegah F, Laakso M. Do articles in open access journals have more frequent altmetric activity than articles in subscription-based journals? An investigation of the research output of Finnish universities. Scientometrics. 2020;122(1):645-59. doi: 10.1007/s11192-019-03301-x.

36. Dehdarirad T, Didegah F. To What Extent Does the Open Access Status of Articles Predict Their Social Media Visibility? A Case Study of Life Sciences and Biomedicine. Journal of Altmetrics 2020;3(1). doi: http://doi.org/10.29024/joa.29.

#The method needs more details and clarification. For instance, it says tweets for a two year period were collected. For example for papers published in 2014, tweets in 2014 and 2015 were collected. Was the month of publication taken into account in this data collection? If not, a paper published in January 2014 would’ve had two years of tweets in the dataset, while a paper in Dec 2014, would have only 13 months’ worth of tweets. The same goes for citation data. 

• Thanks for this comment. In this, paper the number of citations and tweets were aggregated by year and not by month of publications. In bibliometrics it is common to set a time citation window of 3 or 5 years based on the year of publications and not month of publications. In fact, citation window is defined as: the years after publication that is used to count the citations (Bibliometric handbook for Karolinska institutet, 2014) https://kib.ki.se/sites/default/files/bibliometric_handbook_2014.pdf. So, it refers to years after publication. However, following the reviewer’s comment, to make it clearer and to be precise, I have changed ‘after publication’ to ‘after publication year’ (page 6 lines 122-125). I also replaced ‘at least three years’ with ‘a time citation of window of at least three years.’ 

#How many authors (first and last) were there in the dataset?

• Regarding this comment, it should be considered that the analysis in this paper is based on authorship positions and not by the number of publications for unique authors in each authorship position. In other words, the analysis in the paper is at paper level and not author level. The data set of this study was comprised 47,961 papers. So, there are 47,961 authorship positions for both last and first authorship position. The gender of authors in 12 authorship positions were not detected as mentioned on page line 7, line 145. However, to be more precise, I have added this line to page 7, lines 147-148: ‘These 12 authorships accounted for seven first and five last authorship positions.’ Thus, in this paper, the statistics on the number of authorship positions by each gender is reported in Figure 2 at paper level. 

# and how the publications, citation, self-citation data was obtain'd? Did the author manually search each of those probable thousands of authors? Was there any problem with author disambiguation?

• Thanks for bringing this to my attention. I used Scival API (https://dev.elsevier.com/documentation/SciValAuthorAPI.wadl) to automatically download these three values for each author using their author IDs. I have now corrected this in the manuscript (pages 7, lines 153-159). After correspondence with Clarivate, I realized that currently it is not possible to download the total number of self-citations for each author, automatically. Furthermore, there was not a straightforward way to download the total number of publications and citations for each author using their APIs. So, instead, I used Scival API which allowed me to download these three former-mentioned indicators automatically. For details regarding the steps taken to do this, please see pages 7, lines 153-159 in the manuscript. Finally, in order to address the issue of disambiguation, I used author IDs. In Author history provided by Author API, it was also possible to check and track the affiliation history of authors. 

#Title length: were words like the, a, an, on, and …counted?

• Thanks for this comment. I calculated the title length based on the number of characters and not words in the title of an article. So, I have corrected it in the paper (page 8, line 167). This methodology was used in other bibliometrics studies such as: 

Haustein S, Costas R, Larivière V (2015) Characterizing Social Media Metrics of Scholarly Papers: The Effect of Document Properties and Collaboration Patterns. PLOS ONE 10(3): e0120495. https://doi.org/10.1371/journal.pone.0120495

Zahedi Z, Haustein S. On the relationships between bibliographic characteristics of scientific documents and citation and Mendeley readership counts: A large-scale analysis of Web of Science publications. Journal of Informetrics. 2018;12(1):191-202. doi: https://doi.org/10.1016/j.joi.2017.12.005.

# Abstract readability, how was it calculated? Did software (text processing) do this or somebody had to read all of the abstracts and assign a score? How about the validity and reliability issues here?

• Quanteda R Package was used to calculate the score for each abstract. So, it is not done manually. I used textstat_readability command which returns a data.frame of documents and their readability scores. More information on this package can be found at this link: https://rdrr.io/cran/quanteda/man/textstat_readability.html.

• On page 8 line 186, it is now mentioned that this was done using quanteda R package. 

#Figure 1 should have proper legends with values shown on the bars (e.g. percentage).

• Following the reviewer’s comment, in Figure 1, the percentage % sign has been added to the values in the Y axis. The values were also shown in percentages on the bars. Figure 1 is now Figure 2 in the manuscript.

#The paper needs a table that presents some descriptive statistics about the variables included in the study. For instance, how many authors, how many papers from each subject category, what was the average and median title length, how many OA and non-OA, how many papers had funding and how many didn’t, average, mean of the number of authors etc.

• Thanks for this comment. Following the reviewer’s comment, Table 3 has been added to page 11, which provides descriptive statistics at paper level for the covariates entered in regression models 1 and 3. 

#I believe the level of accuracy used in the paper for significance reporting (shown with long exponents, e.g. 2.45e-05) is unnecessary, up to 3 decimal points would suffice.

• They have now been modified. 

#Also I think the author needs to make the contribution clear in the paper given the focus is on association of tweets and citation (adding gender to the issue) and there has already been some good research on that.

• It has now been modified in the manuscript.

#Language, proofreading will improve the paper. It seems the paper has one author, but throughout the paper, the author uses 'we' to present the study which might not be right.

• Thanks for this comment. It has been corrected. 

#Typo: p. 209, line 206, as well as well as higher

• It has been corrected. 

#Typo, p. 16, line 299, cations

• It has been corrected.

---

## [Decision Letter · Decision Letter 1]

20 Oct 2020

Could early tweet counts predict later citation counts? A gender study in Life Sciences and Biomedicine (2014-2016)

PONE-D-20-22052R1

Dear Dr. Dehdarirad,

We’re pleased to inform you that your manuscript has been judged scientifically suitable for publication and will be formally accepted for publication once it meets all outstanding technical requirements.

Kind regards,

Alireza Abbasi

Academic Editor

PLOS ONE

Additional Editor Comments (optional):

Thank you for the revision. We noticed almost all teh comments are address properly. However, we advise to highlight the contribution of the paper (as advised by a reviewer as well) perhaps in the Abstract or Introduction, and Conclusion. 

Reviewers' comments:

Reviewer's Responses to Questions

**Comments to the Author**

1. If the authors have adequately addressed your comments raised in a previous round of review and you feel that this manuscript is now acceptable for publication, you may indicate that here to bypass the “Comments to the Author” section, enter your conflict of interest statement in the “Confidential to Editor” section, and submit your "Accept" recommendation.

Reviewer #2: All comments have been addressed

2. Is the manuscript technically sound, and do the data support the conclusions?

Reviewer #2: Yes

3. Has the statistical analysis been performed appropriately and rigorously? 

Reviewer #2: Yes

4. Have the authors made all data underlying the findings in their manuscript fully available?

Reviewer #2: Yes

5. Is the manuscript presented in an intelligible fashion and written in standard English?

Reviewer #2: Yes

6. Review Comments to the Author

Reviewer #2: The author has adequately addressed all of the comments. The paper has been improved and the method has been explained more thoroughly.

7. PLOS authors have the option to publish the peer review history of their article (what does this mean?). If published, this will include your full peer review and any attached files.

Reviewer #2: **Yes: **Hamid R. Jamali

---

## [Editor Report · Acceptance letter]

22 Oct 2020

PONE-D-20-22052R1 

Could early tweet counts predict later citation counts? A gender study in Life Sciences and Biomedicine (2014-2016) 

Dear Dr. Dehdarirad:

I'm pleased to inform you that your manuscript has been deemed suitable for publication in PLOS ONE. Congratulations! Your manuscript is now with our production department. 

Kind regards, 

on behalf of

Dr. Alireza Abbasi 

Academic Editor

PLOS ONE